# Negative Regulation of ULK1 by microRNA-106a in Autophagy Induced by a Triple Drug Combination in Colorectal Cancer Cells *In Vitro*

**DOI:** 10.3390/genes12020245

**Published:** 2021-02-09

**Authors:** Rebeca Salgado-García, Jossimar Coronel-Hernández, Izamary Delgado-Waldo, David Cantú de León, Verónica García-Castillo, Eduardo López-Urrutia, Ma. Concepción Gutiérrez-Ruiz, Carlos Pérez-Plasencia, Nadia Jacobo-Herrera

**Affiliations:** 1Postgraduate in Experimental Biology, DCBS, Autonomous Metropolitan University-Iztapalapa, Iztapalapa, Mexico City C.P. 09340, Mexico; rebe_zoid92@hotmail.com; 2Genomics Laboratory, The National Cancer Institute, Tlalpan, Mexico City C.P. 14080, Mexico; jossi_thunders@hotmail.com (J.C.-H.); izz.waldo11@gmail.com (I.D.-W.); 3Functional Genomics Laboratory, Biomedicine Unit, FES-IZTACALA, UNAM, Tlalnepantla C.P. 54090, Mexico; garciaver@gmail.com (V.G.-C.); e_urrutia@unam.mx (E.L.-U.); 4Cancer Biomedical Research Unit, Genomics Laboratory, National Cancer Institute, Tlalpan, Mexico City C.P. 14080, Mexico; dfcantu@gmail.com; 5Laboratory of Experimental Medicine, Translational Medicine Unit, Institute of Bio-Medical Research, UNAM/National Institute of Cardiology Ignacio Chavez, Tlalpan, Mexico City C.P. 14080, Mexico; mcgr@xanum.uam.mx; 6Department of Health Sciences, Autonomous Metropolitan University-Iztapalapa, Iztapalapa, Mexico City C.P. 09340, Mexico; 7Biochemistry Unit, Institute of Medical Sciences and Nutrition, Salvador Zubirán, Tlalpan, Mexico City C.P. 14080, Mexico

**Keywords:** autophagy, miR-106a, ULK1, colorectal cancer, HCT116, SW480, metformin, doxorubicin, sodium oxamate

## Abstract

Colorectal cancer (CRC) is among the top three most deadly cancers worldwide. The survival rate for this disease has not been reduced despite the treatments, the reason why the search for therapeutic alternatives continues to be a priority issue in oncology. In this research work, we tested our successful pharmacological combination of three drugs, metformin, doxorubicin, and sodium oxamate (triple therapy, or TT), as an autophagy inducer. Firstly, we employed western blot (WB) assays, where we observed that after 8 h of stimulation with TT, the proteins Unc-51 like autophagy activating kinase 1(ULK1), becline-1, autophagy related 1 protein (Atg4), and LC3 increased in the CRC cell lines HCT116 and SW480 in contrast to monotherapy with doxorubicin. The overexpression of these proteins indicated the beginning of autophagy flow through the activation of ULK1 and the hyperlipidation of LC3 at the beginning of this process. Moreover, we confirm that ULK1 is a bona fide target of hsa-miR-106a-5p (referred to from here on as miR-106a) in HCT116. We also observed through the GFP-LC3 fusion protein that in the presence of miR-106a, the accumulation of autophagy vesicles in cells stimulated with TT is inhibited. These results show that the TT triggered autophagy to modulate miR-106a/ULK1 expression, probably affecting different cellular pathways involved in cellular proliferation, survivance, metabolic maintenance, and cell death. Therefore, considering the importance of autophagy in cancer biology, the study of miRNAs that regulate autophagy in cancer will allow a better understanding of malignant tumors and lead to the development of new disease markers and therapeutic strategies.

## 1. Introduction

Colorectal cancer (CRC) is the third most common neoplasia around the world for both sexes [1]. The last report of the International Agency for Research of Cancer (2018) stated that this disease is responsible for 9.2% of cancer deaths [2]. Clinically, CRC is staged in three categories: localized, regional, and distant or metastatic [3]. The standard drug options are 5-fluorouracil (FU)/leucovorin/capecitabine, capecitabine/oxaliplatin, or 5-FU/oxaliplatin, depending mainly on the patient’s condition [4,5]. Despite surgery or chemotherapy based on cytotoxic or targeted therapy, the overall survival rate for metastatic CRC is about 14%, revealing the need to distinguish new therapeutic targets [6]. In this regard, the role of autophagy in cancer has motivated the interest of the scientific community in identifying the mechanisms that participate in the regulation of this intricate process.

Autophagy is a highly conserved process that maintains cellular homeostasis [7]. The autophagy flow is initiated in response to a stress stimulus, such as nutrient deprivation, amino acid starvation, and hypoxia, among others [8]. Thus cell survival is prolonged by the degradation of macromolecules and organelles through lysosomal degeneration [9]. In cancer, autophagy has a dual function. Some authors postulate that autophagy is activated as a survival mechanism to avoid cytotoxicity caused by antineoplastic drugs [10,11]. On the other hand, our group showed that prolonged autophagy could lead to apoptosis in cancer cells [12]. Although the activation of autophagy can support cell survival by contributing to treatment resistance, the use of drugs that block glycolysis and the mTOR pathway could activate autophagy in a persistent way, leading to the apoptosis of tumor cells [13,14].

Different signaling pathways regulating autophagic flux have been identified; however, the fine regulation mechanisms exerted by microRNAs are little characterized [15]. MicroRNAs, or miRNAs, are small non-codification RNAs made of 20 to 25 nucleotides [16]. miRNAs work as negative regulators of genic expression, controlling several biological processes in animals, such as cellular differentiation and proliferation, apoptosis, stem cell division, and autophagy [17]. Such processes are deregulated in cancer, suggesting that the failure of the interactions of miRNAs with oncogenes or tumor suppressors could interfere in tumorigenesis [18]. In autophagy, miRNAs are involved in different steps of the process, which could be either in the induction, in the formation of nuclear vesicles, or in the ending stages, such as elongation and vesicle finishing [19]. Furthermore, miRNAs can regulate signaling pathways implicated in the mammalian target of rapamycin (mTOR), ULK1 kinase complex, becline-1, and ATG4 signaling, among others [20,21]. 

miRNA-106a is an oncogenic miRNA that participates in the early stages of autophagy [22]. It targets ULK1, which once inhibited blocks the induction of autophagy [23,24]. Other targets for miR-106a are the Atg13 and FAK family, which belong to the ULK1 complex, also involved in autophagy induction. In colon cancer, miR-106a participates in the progression of the disease and its expression correlates with the presence of the tumor, suggesting miR-106a as a diagnostic molecular biomarker [25]. Our research group proved that the pharmacological combination of metformin-sodium oxamate and doxorubicin induces cell death by apoptosis throughout the maintenance of autophagic flow by the hyperlipidation of LC3 and the inactivation of mTOR [12,26]. In this regard, this study aimed to identify the role of miR-106a in the regulation of pharmacologically induced autophagy in CRC.

To accomplish this objective, we analyzed the role of miR-106a in the regulation of autophagy in CRC tumor cells. Our results confirm that ULK1 is a bona fide target of miR-106a when a pharmacological combination (triple therapy (TT)) is used that produces sustained autophagy and directs the cells to apoptosis. For this purpose, we examined the initiation of different early markers of autophagy in cells treated with the previously described pharmacological combination in a monolayer CRC cell culture. Subsequently, we proved that miR-106a binds to the 3’ UTR region of ULK1 and induces the inhibition of protein synthesis. Finally, using the GFP-LC3 fusion protein, we observed that in the presence of miR-106a, the accumulation of autophagic vesicles in the cells treated with the drug combination is inhibited. Our results confirm that the drug combination induces autophagic flow. Furthermore, it can inhibit, by still undescribed mechanisms, the expression of negative regulators, such as miR-106a, in CRC cells.

## 2. Materials and Methods

### 2.1. Cell Culture, Drugs, and Transfection

CRC-derived HCT116 and SW480 colorectal cancer cells (American Type Culture Collection ATCC CCL-247 and CCL-228) were maintained in RPMI medium and McCoy’s medium, respectively, supplemented with 10% fetal bovie serum (FBS) (Corning, New York, NY, USA) and cultured at 37 °C in an atmosphere of 5% CO_2_. Non-tumoral immortalized epithelial CRL1790 colon cells (ATCC CDN 841 CoN) were cultured in DMEM F12 medium (Corning, New York, NY, USA). The cell line HCT116 is a colorectal carcinoma from an adult male, characterized by a mutation in codon 12 of the ras proto-oncogene, while SW480 is a Duke´s type B colorectal adenocarcinoma disease. According to its description on the ATCC page, SW480 expresses the p53 protein and the oncogenes c-myc, K-ras, H-ras, N-ras, myb, sis, and fos [27,28]. The drugs used were doxorubicin (Dox, 0.75 μM) and, for triple therapy, metformin (Met, 12 mM), sodium oxamate (Ox, 10 mM), and doxorubicin (Dox, 0.4 μM), with untreated cells as the negative control (basal condition of the respective cell line). Rapamycin was used as autophagy positive control (10 μM).

All plasmids used for this study were transfected with lipofectamine 2000 transfection agent (Invitrogen, Carlsbad, CA, USA), following the manufacturer’s protocol. MicroRNA mimics and inhibitors were transfected using the siPORT NeoFX transfection agent (Life Technologies, Carlsbad, CA, USA) following the manufacturer’s protocol. For functional assays, we transfected an miR-106a mimic or an miR-106a anti-miRNA, as well as an unrelated miRNA (miR-1) or a scrambled 20-nt sequence, as control (all of them purchased from Ambion, by Life Technologies, Foster City, CA, USA).

### 2.2. Protein Expression Analysis

After treatments, total protein was extracted from cultured cells usingradioimmunoprecipitation assay buffer (RIPA buffer, Santa Cruz Biotechnology sc-24948, Dallas, TX, USA). For immunodetection, 30 µg total protein from cultured cells was mixed with Laemmli sample buffer, boiled, separated in 10% SDS-PAGE for ULK1 and 12% or 15% SDS-PAGE for the others proteins, and transferred in a polyvinylidene difluoride (PVDF membrane, Amersham-GE Healthcare, London, UK) in a semidry chamber trans-blot turbo (Bio-Rad) at 25 V, 1 mA, in 30 min. After blocking with 5% non-fat dry milk for 2 h, the membrane was incubated with the specific antibody overnight at 4 °C using the following dilutions: 1:1000 for anti-ULK1 (cell signaling) and 1:3000 for anti-becline-1 (cell signaling), anti-LC3 (cell signaling), and anti-ATG4b (Santa Cruz Biotechnology, Dallas, TX, USA). For detection, 1:3000 dilutions of HRP anti-rabbit or anti-mouse conjugate antibodies (Santa Cruz Biotechnology, Dallas, TX, USA) were used. Finally, for membrane visualization, the SuperSignal West Femto chemiluminescent substrate (Pierce, Rockford, IL, USA) was used in a C-DiGit™ scanner (LI-COR, Lincoln, NE, USA), employing the IMAGE STUDIO (LI-COR, Nebraska, USA) software. Actin (anti-actin, Sc-47778) was the loading control. Membranes were stripped and re-probed for the detection of actin (anti-actin, Sc-47778) as a loading control. A representative image from four independent experiments is shown. 

### 2.3. TUNEL Assay

Cells at 80% confluency were treated with different combinations of drugs for 24 h. The cells were fixed with 4% paraformaldehyde and permeabilized with 0.2% Triton X-100. DNA fragmentation was determined by TdT-mediated dUTP nick end labeling (TUNEL, Promega Corporation, Madison, WI, USA) as described by the manufacturer (DeadEndTM Fluorometric TUNEL System, Promega Part# TB235). Fluorescent images were obtained using a Leica TCS SP8 (Leica Microsystems, Wetzlar, Germany) inverted microscope (objective 40 ×). All images were processed with Leica Application Suite X and Illustrator CC.

### 2.4. qRT-PCR

Total RNA was purified from cultured cells (CRL1790, HCT116, and SW480) using the TRIzol reagent (Invitrogen, Carlsbad, CA, USA) following the manufacturer’s protocol. The quality and concentration of RNA in samples were determined using gel electrophoresis and the Epoch Microplate Spectrophotometer (Bio-Tek, Tek Instruments, Inc. Winooski, VT, USA). To measure miR-106a, cDNA was generated from 100 ng total RNA with the TaqMan Micro-RNA Reverse Transcription Kit (Applied Biosystems, Foster City, CA, USA) in a 15 µL volume; quantitative polymerase chain reaction (qPCR) was performed using 1 µL cDNA and the miR-106a TaqMan probe with TaqMan Universalpolymerase chain reaction (PCR) Master Mix 2. Amplification conditions were 10 min at 95 °C, followed by 40 cycles at 95 °C for 15 s and 68 °C for 60 s. For ULK1 mRNA detection, cDNA was synthesized from 2 µg total RNA using the High-Capacity cDNA Reverse Transcription Kit (Applied Biosystems); 1 µl of this reaction was used for qPCR. Amplification conditions were initial denaturation for 2 min at 95 °C, followed by 40 cycles at 95 °C for 15 s, primer-dependent annealing temperature for 15 s, and 72 °C for 60 s. Relative expression data were calculated through the ΔΔCt method (Applied Biosystems) and normalized relative to U6 snRNA or GAPDH mRNA accordingly.

### 2.5. Luciferase Reporter Assays

Reporter plasmids were constructed by ligation of synthetic oligonucleotide duplexes (IDT) containing putative miR-106a target regions in the ULK1 3′ UTR, 5′ CTAGTTTGTCAATCACCCAAGCACTTTAA3′ and 3′AAACAGTTAGTGGGTTCGTGAAATTTCGA 5′, obtained from microRNA.org, to form a DNA duplex with overhanging SpeI and HindIII half-sites at the 5′ and 3′ ends, respectively, which was cloned into the digested p-MIR-Report plasmid (Ambion). For the mutated-type construct, the mutant 3′ UTR sequence of the target gene was synthesized and inserted into the p-MIR-Report vector. These constructs were co-transfected with miR-106a mirVana miRNA mimic (Applied Biosystems) and the pRL Renilla Luciferase Control Reporter (Promega, Corporation, Madison, WI, USA) into HCT116 and SW480 cells. Luciferase activity was analyzed using the Dual-Luciferase Reporter Assay System (Promega Corporation, Madison, WI, USA) 24 h after transfection, in a GloMax 96 Microplate Luminometer (Promega). Luciferase activity was normalized to Renilla activity for each transfected well; each experiment was performed in triplicate.

### 2.6. Monitoring Autophagy by the Formation of GFP-LC3 Puncta

For autophagy assay, HCT116 and SW480 cells were seeded in six-well plates with coverslips for 24 h. The next day, cells expressing GFP-LC3 (autophagy marker) were treated with 2% FBS and transfected with the mimics of miR-106a, anti-miR, miR-1, and scramble. The cells were monitored at 24 h of treatment. As a positive control were included cells treated only with lipofectamine. Later, they were fixed using 3.7% paraformaldehyde (PFA) at room temperature for 30 min and permeabilized with 0.5% Triton X-100 for 3 min. Then, coverslips were rinsed with PBS and mounted with Vectashield (Vector Laboratories VECTASHIELD® Antifade Mounting Medium with DAPI) and fluorescent images were acquired using the Leica TCS SP8 inverted microscope (objective 40 ×). All images were processed with Leica Application Suite X.

### 2.7. Statistical Analysis

All values are expressed as the mean ± SEM. Data were analyzed using a one-way ANOVA analysis followed by Tukey’s multiple comparison test. For all statistical analyses, we used the GraphPad PRISM version 5.0 software.

## 3. Results

### 3.1. Triple Therapy Promotes Autophagy in CRC Cells

Our research group previously showed that TT promotes autophagy in both breast cancer and CRC in vivo and in vitro [12,26]. We also demonstrated that sustained autophagy leads to tumor cell death by apoptosis. In this work, we investigated if the autophagic flow was activated when CRC cells were stimulated with the aforementioned drug combination. Thus, we searched for the effect of monotherapy (doxorubicin), the combination of Met/Ox, and TT on canonical autophagy markers in HCT116, SW480, and CRL1790. Cells treated with doxorubicin showed an increase in ULK1, becline-1, ATG4, and LC3 I/LC3 II ratio after two hours of treatment, suggesting that autophagy was taking place. However, after six hours of drug application, this effect was not observed in both cell lines (Figure 1a,c). On the other hand, the TT induced the selected autophagy canonical markers for four hours after the end of treatment (Figure 1b,d). The conversion of the LC3 protein from LC3 I to LC3 II was induced. The LC3 I/LC3 II ratio was clearly augmented after treatment with the triple pharmacological combination. Notoriously, ULK1 detection increased gradually over the course of treatment (Figure 1b,d). Such findings are supported by the densitometry results; the difference in protein expression is evident between TT and doxorubicin (Appendix A). Meanwhile, the combination of Met/Ox slightly maintained the expression of ULK1, becline-1, and ATG4. However, the LC3 I/LC3 II ratio was barely visible in Met/Ox, indicating that autophagy flux may not be initiated or it is happening only in the early stages of autophagy (Appendix A). It was the same for HCT116 and SW480 cells. Thus, these data confirm that TT enhances the detection of canonical markers of autophagy more effectively than the rapamycin control. Moreover, to demonstrate that our pharmacological combination produces not only autophagy but also apoptosis, we carried out the TUNEL assay (Figure 2 and Appendix A). The photos in Figure 2a,c show that TT promoted apoptosis in both cell lines, compared to Dox, which had a minimal apoptotic effect. Besides, by Western blot different protein markers involved in the apoptosis process were detected. As depicted in Figure 2b,d, apoptotic cell death is detected by the cleavage of caspase 3 in TT-treated cells compared to nontreated cells, suggesting extrinsic apoptosis pathway activation. In summary, Figure 1 and Figure 2 indicate that TT induces autophagy and apoptosis in HCT116 and SW480 cells.

### 3.2. ULK1 Is Negatively Regulated by miR-106a in HCT116 CRC Cells

ULK1 plays a key role in the formation of autophagophores, the precursors of autophagosomes [29]. The results of the immunodetection show that ULK1 increases progressively in a time-dependent manner. So, we hypothesized that there was a post-transcriptional regulator that could finely modulate the expression of ULK1. Therefore, we examined in bioinformatics databases the best miRNA candidates that could interact with ULK1 3′ UTR using the algorithms of TargetScan [30], MirDataBase [31], and Starbase [32]. After the search was done, we identified miR-106a as a putative candidate that could target ULK1. Thus, we analyzed miR-106a expression along with the pharmacological treatments. Dox did not change miR-106a expression during the first four hours of drug exposition, but it had a significant increase (*p* < 0.05) at six hours (Figure 1e,g), correlating with the reduction of ULK1 protein (Figure 1a,c). Notably, TT steadily weakened miR-106a expression or presumably abolished it at six hours in SW480 and HCT116, respectively (Figure 1f,h). Conversely, ULK1 exhibited a gradual increase (Figure 1b,d, showing an inverse correlation with miR-106a expression). All PCR miR-106a expression results were compared to the basal condition, or Time 0h (Figure 1e–h). On the other hand, as a reference, in Appendix A are the expression values of miR-106a when cells are treated with the duplet Met/Ox. As depicted in Appendix A, HCT116 cells treated with Met/Ox showed an abrupt decrease in miR-106a (*p* < 0.001) at all hours, while the expression of ULK1 remained steady but with no change in the same experimental conditions (Appendix A). Meanwhile, miR-106a expression in SW480 showed a marked decrease at 4 h after treatment but an increase at 6 h (Appendix A). However, ULK1 expression is similar in both cell lines from 8 h to 24 h after drug stimulation (Appendix A).

To check whether miR-106a exerted direct regulation on the ULK1 messenger, we performed luciferase reporter assays in HCT116 and SW480 cells transfected with an miR-106a mimic or the corresponding anti-miRNA. The luciferase reporter system was designed with the 3′ UTR-specific interaction region of ULK1/miR-106a (Figure 3a) that was cloned upstream of luciferase ORF in the p-MIR-Report plasmid. This construct was called p-ULK1 and its mutated version p-ULK1-mut (matched control). In HCT116 cells, the luciferase expression was significantly decreased when cells were co-transfected with the miR-106a and interacting region% (*p* < 0.001) (Figure 3b); as expected, the mutated ULK region did not reduce the level of luciferase expression. However, the transfection with anti-miR-106a recovered the luminescence emission, demonstrating a bona fide interaction between miR-106a and the 3′ UTR region of ULK1. Similarly, the different negative controls did not cut the expression of luciferase. This result was consistent when analyzing the presence of ULK1 by Western blot (Figure 3c). Transfection with miR-106a inhibited the expression of the protein, but the presence of an anti-miR-106a rescued the ULK1 protein expression, suggesting that there is a post-transcriptional interaction between both genes and that it affects the protein levels of ULK1. Interestingly, this phenomenon was not the same for SW480. As clearly seen in Figure 3d,e, luminescence was not reduced in co-transfected cells, suggesting that the effect is cellular context dependent. Even when both cells lines are quite similar, the autophagy process is regulated by another miRNA. In Appendix A, we observed the successful transfection of miR-106a and controls in HCT116 and SW480. Even more, we measured the miR-106a expression by qPCR in HCT116, SW480, and CRL1790 CRC cell lines and non-tumor cells (Appendix A). As hypothesized, miR-106a was overexpressed in the tumor cell lines (Appendix A). HCT116 had a six-fold overexpression, and SW480 had a four-fold overexpression. Besides, we analyzed the expression levels of mRNA ULK1, proposed as a direct target of miR-106a, and found no significant changes of ULK1 in HCT116 similarly to the non-tumor cell line. However, in the metastatic cell line (SW480), ULK1 expression decreased significantly (Appendix A).

### 3.3. miR-106a Inhibition Induces the Formation of GFP-LC3 Puncta in HCT116 and SW480 Cells

So far, our findings indicate that miR-106a regulates autophagy through inhibition of ULK1 in HCT116. Nevertheless, we performed the same experiment in both cell lines. As a conclusive experiment of the participation of this microRNA in the regulation of autophagy, we employed the formation of GFP-LC3 puncta in the presence or absence of miR-106a. Consequently, cells were induced into serum starvation through cultivation with 2% FBS. Figure 4 shows in green the accumulation of GFP-LC3 fusion protein when autophagy is induced by triple therapy. When transfected with the mimic of miR-106a, a reduction of GFP-LC3 puncta in the cytoplasm is observed. The re-establishment of autophagy denoted by the formation of GFP-LC3 puncta is visible when the presence of miR-106a is inhibited by employing a sequence capable of inhibiting its function (GFP-LC3/anti-miR-106a). These results confirm the involvement of miR-106a in the regulation of the autophagic flow in CRC cells. The lipofectamine control indicates that autophagy was not affected by the reaction.

## 4. Discussion

Colorectal cancer therapy is still a challenge due to the complexity and molecular variety of this disease [33]. Current standard treatments include 5-fluorouracil and oxaliplatin, where chemoresistance is the main reason for these treatments’ failure [34]. Our group reported a successful pharmacological combination that abolishes the cancer’s progression in a colitis-associated cancer azoxymethane (AOM)/dextran sodium sulfate (DSS) mouse model [26] by autophagy and apoptosis induction, giving place to a new, promising therapy in CRC. However, the molecular mechanism of how the aforementioned pharmacological combination promotes apoptosis/autophagy remains unclear. In this paper, we looked deeper to understand the role of TT in autophagy, evaluating the miR-106a-level expression in HCT116 and SW480 cells. We observed a full decline in miR-106a expression in the TT group as well as apoptosis (analyzed by TUNEL assay) and autophagy induction via ULK1/miR-106a regulation in HCT116. Our rationale behind this research is established on previous reports where important overexpression of miR-106a in colorectal tumor tissues is shown compared to normal mucosa [35].

The TT formulation includes the drugs doxorubicin, metformin, and sodium oxamate. The basis for the selection of these three drugs is their versatile mechanisms of action and their ability to cover different hallmarks of the cancer cell simultaneously. Briefly, doxorubicin is one of the main drugs used in chemotherapy; it promotes cell death by DNA damage, oxidative stress, and mitochondrial dysfunction [36]. On the other hand, metformin inhibits complex 1 of the electron transport chain in the mitochondria, activating AMPK and abrogating mTOR phosphorylation, leading to autophagy activation [37]. Together with this fact, metformin is considered a powerful hypoglycemic that reduces glucose blood levels, promoting a decrease in the metabolic rate [38]. Otherwise, sodium oxamate, an inhibitor of lactate dehydrogenase A (LDHA1), induces cell death in several cancers [39,40,41]. In this investigation, we confirmed the ability of TT to induce autophagy in two different colon cancer cells even better than monotherapy (doxorubicin) (Figure 1) and duplet (Met/Ox) (Appendix A). For instance, in cells administered with Dox, the LC3 II (key autophagy marker) was activated in the first two hours of treatment, but this effect finished up to six hours after treatment. In our previous report [26], LC3 II was activated 24 h after the stimulus. Therefore, we could suggest that Dox does not sustain the autophagic flux properly. The duplet Met/Ox improved autophagy induction (sustained expression of ULK1 and LC3 II) compared with Dox alone (Figure 1a,c; Appendix A), but the most notable autophagy effect was with the combination of the three drugs.

Several studies have demonstrated that microRNAs regulate critical functions like cellular proliferation and cell death [42,43,44]. MicroRNAs modulate autophagy through their effects on various autophagy regulatory proteins, such as ULK1, the principal inducer of this process, which has been identified as a direct target of different microRNAs in various types of cancer [45]. For instance, miR-25 is a regulator of autophagy and cell death through its direct effects on the expression of ULK1 in breast cancer [46]. Furthermore, it was observed that direct interactions among miR-20A and miR-106b and ULK1 could lead to the inhibition of autophagy induced by leucine deprivation in C2C12 myoblast cells while blocking miR-20a and endogenous miR-106b could restore normal autophagic activity [47]. Moreover, Huang et al. (2011) observed that in squamous carcinoma cells, cisplatin-induced miR-885-3p directly targets ULK1 and contributes to autophagy regulation [48]. In this way, pharmacological treatments modulate proliferation and apoptosis by regulating the global microRNA expression profile [49].

To the best of our knowledge, a single microRNA can negatively regulate hundreds of mRNAs. We are sure of the necessity to demonstrate the negative regulation of a single microRNA in a target gene to know the role of microRNAs in such regulation. Although they are reductionist approaches, they are needed to validate the function of specific microRNAs in different aspects of cancer biology. In the case of the work presented here, a goal arising from the treatment with the triple therapy was the reduction of ULK1. A plausible hypothesis for the reduction was the participation of a microRNA. Certainly, miR-106a negatively regulates different genes in the tested cell lines, but it is not possible, from our perspective, to assure that the role of this microRNA is only the regulation of ULK1. 

Our findings suggest that TT decreases miR-106a expression in CRC cells (Figure 1f), which leads to the induction of the autophagy process and subsequently apoptosis. As known, ULK1 is a protein that participates in the initiation of autophagy and its downregulation via microRNAs could inhibit autophagy progression in many types of cancers [50,51]. We observed that ULK1 is a direct target of miR-106a in HCT116 (Figure 3b), which when bound to ULK1, inhibits its expression, causing the repression of autophagy. Finally, to confirm the participation of miR-106a in autophagy, the GFP-LC3 experiment was carried out (Figure 4). We observed a decrease in this process in the presence of miR-106a. However, in the presence of the antimiR, autophagy was restored. In lung cancer, it was reported that miR-106a can negatively regulate ULK1 by inhibiting autophagy. This mechanism was analyzed in the presence of tyrosine kinase inhibitors [52]. Thus, our study is in agreement with the findings in lung cancer demonstrating the participation of miR-106a in the negative regulation of autophagy in HCT116 CRC cells.

There are a large number of investigations that show autophagy as a mechanism that confers protection against the induction of cell death by different antineoplastic agents [53,54], radiotherapy [55], or endocrine therapy in breast cancer [56,57] to mention a few examples. Despite the discussion, the role of autophagy in cancer is still debatable. Our previous works on breast cancer and CRC in vitro and in murine models [12,26] in addition to the results presented here suggest that the extended pharmacological promotion of autophagy produced by TT leads tumor cells to apoptosis. Finally, based on all these findings, the mechanism proposed in this work is based on the fact that TT decreases the expression of miR-106a, which leads to the induction of the autophagy flux that is observed in the increased detection of the main markers of this process. Likewise, ULK1 is the main promoter of autophagy, and we observed that this is a direct target of miR-106a, which on binding with ULK1 suppresses its expression, causing the inhibition of autophagy. However, when cells are treated with TT, the expression of miR-106a drops, ULK1 is expressed, and thus the autophagy process can be carried out (Graphical Abstract). Nevertheless, this work is not conclusive for SW480. Even though the autophagy process was initiated by TT and cells died due to apoptosis induction, it was evidenced that ULK1 was not a target of miR-106a. It is worthy to continue investigating the actors involved in this cell line.

Currently, the search for antineoplastic drugs is focused on specific mechanisms that are hallmarks of cancer. The present work shows that the pharmacological induction of autophagy in colorectal-cancer-derived cells promotes the reduction of the levels of miR-106, a microRNA involved in the inhibition of autophagy. While the involvement of TT in promoting autophagy through the reduction of miR-106a levels is clear, it opens up new questions that are worthy of further investigation.

## Figures and Tables

**Figure 1 genes-12-00245-f001:**
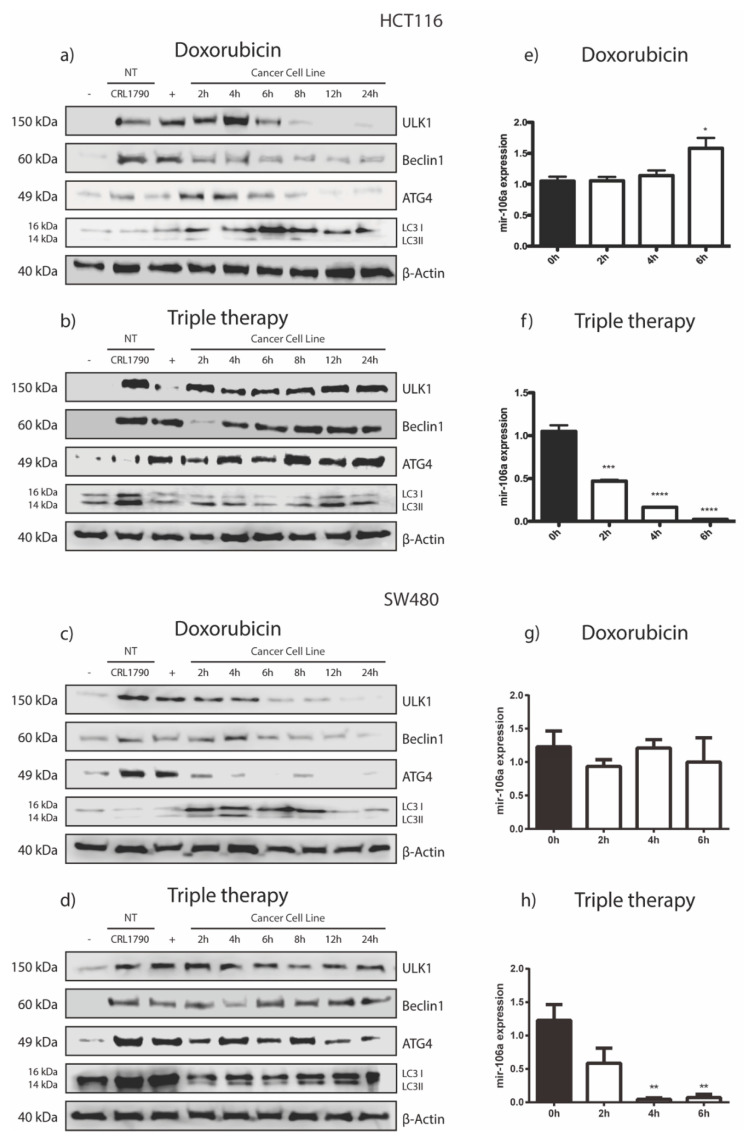
Detection of the proteins involved in the autophagy process in CRL1790 (non-tumor cell line), HCT116, and SW480 colon cancer cell lines. Untreated cells were the negative control, and rapamycin was the positive control (10 μM). Western blot analysis was performed to measure autophagic flux at 2, 4, 6, 8, 12, and 24 h of exposure; (**a**,**c**) doxorubicin; (**b**,**d**) triple therapy. β-actin was used as the loading control. Correlation of miR-106a expression with the protein level expression of ULK1 in the HCT116 and SW480 cell lines. Quantitative real-time PCR shows a slight increase in miR-106a on exposure to doxorubicin (**e**,**g**); and a strong decline was observed in miR-106a on exposure to triple therapy (**f**,**h**). MiR-106a levels were calculated through the ΔΔCt method and normalized relative to U6 snRNA. Data are presented as the mean ± SD of three independent experiments. * *p* < 0.05, ** *p* < 0.01, *** *p* < 0.001, and **** *p* < 0.0001. NT: non-tumor cells.

**Figure 2 genes-12-00245-f002:**
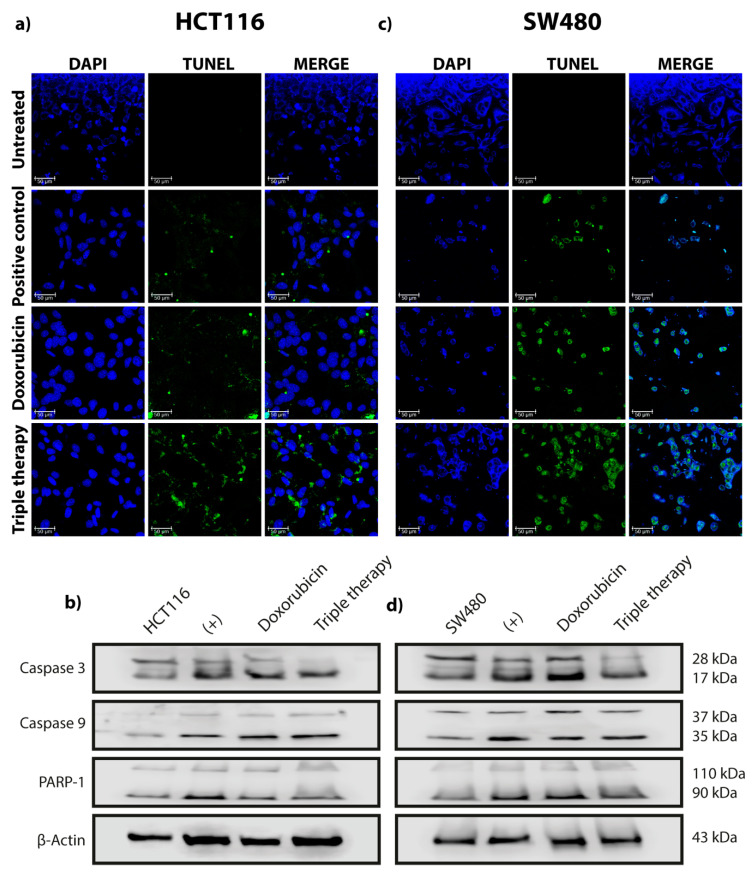
Triple therapy induces cell death by apoptosis in HCT116 and SW480 CRC cells. (**a**,**c**) TUNEL assay to detect HCT116 and SW480 apoptotic cells treated with doxorubicin and triple therapy for 24 h. Green dots indicate apoptosis in each cell, and the color blue corresponds to DAPI stained for nuclei. Images were obtained by confocal microscopy objective 40× (Leica TCS-SP8, Wetzlar, Germany); the experiment was done in duplicate. Scale bar of 50 µm. (**b**,**d**) Detection of the proteins involved in the apoptosis process by Western blot at 24 h of exposure to doxorubicin and triple therapy in the HCT116 and SW480 cell lines, respectively.

**Figure 3 genes-12-00245-f003:**
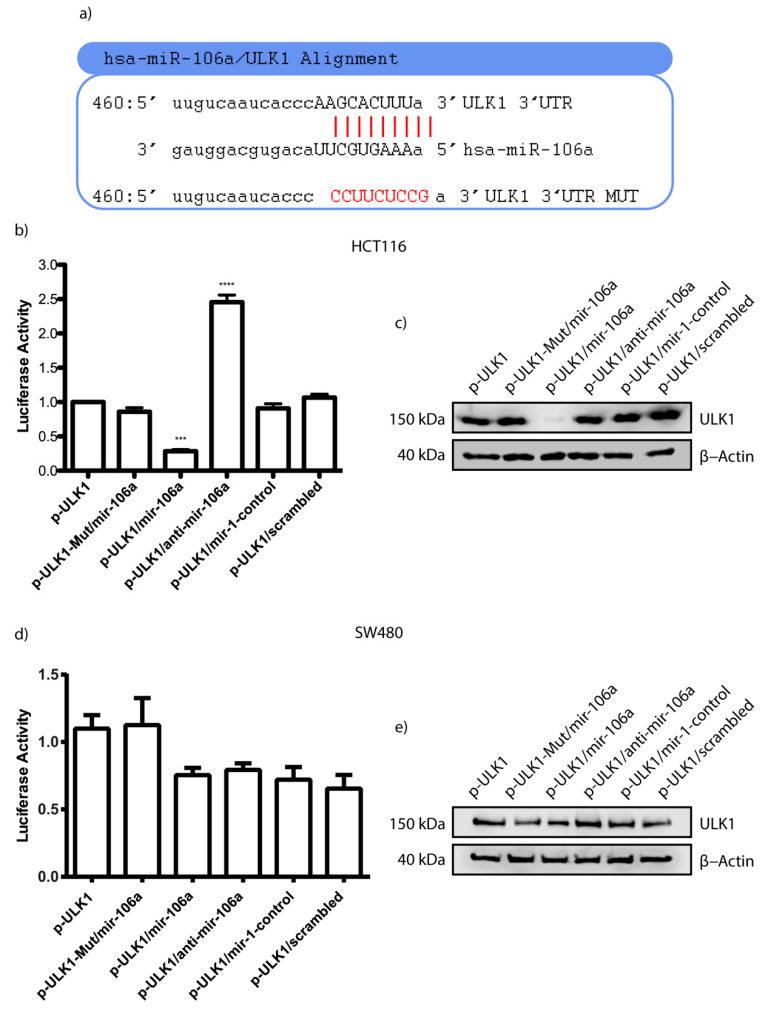
Validation of miR-106a target by Luciferase assay. (**a**) Bioinformatics analyses predicted the binding sites between ULK1 and miR-106a. (**b**,**d**) Luciferase reporter assay was performed to demonstrate the relationship between ULK1 and miR-106a in HCT116 and SW480 cells using the miR-106a/ULK1 interaction region normalized with an empty vector. (**c**,**e**) Detection of the ULK1 protein in the presence of miR-106a. Data are reported as the mean ± SD of three independent experiments. *** *p* < 0.001; **** *p* < 0.0001.

**Figure 4 genes-12-00245-f004:**
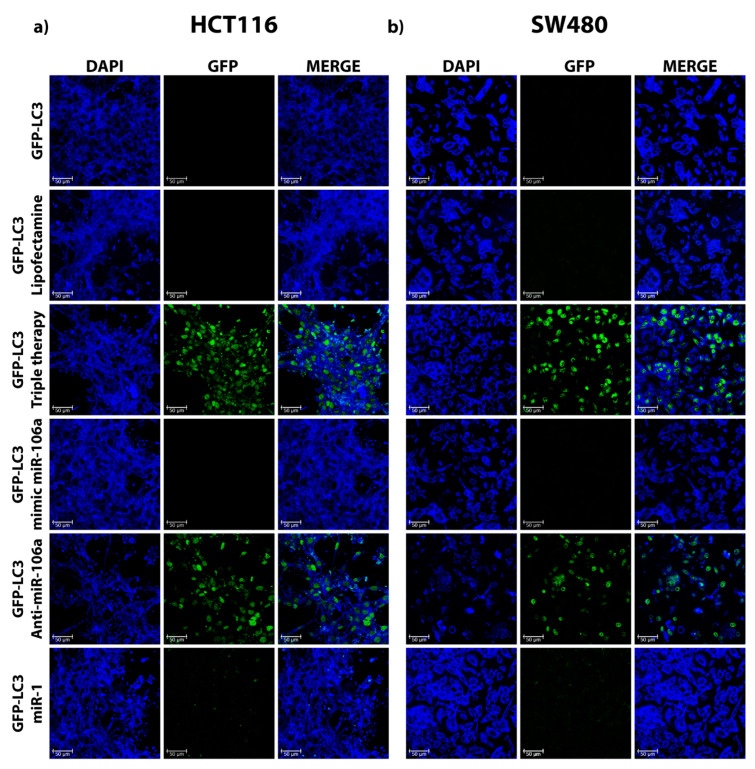
Autophagosome formation in colon cancer cell lines. (**a,b**) HCT116 and SW480 CRC cells were treated under different conditions, and they were transfected with LC3-GFP. Effects of lipofectamine, triple therapy, mimic miR-106a, and antimiR-106a on autophagy flux were observed by confocal microscopy. Images were obtained by confocal microscopy objective 40× (Leica TCS-SP8, Germany); the experiment was done in duplicate. Scale bar of 50 µm.

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
