# Peer review of "Negative Regulation of ULK1 by microRNA-106a in Autophagy Induced by a Triple Drug Combination in Colorectal Cancer Cells In Vitro"

_genes, 2021, doi:10.3390/genes12020245_

Round 1
Reviewer 1 Report
In this work Garcia et al. present an interesting work demonstrating the use of triple therapy to induce autophagy in Colorectal cancer with an additional layer of regulation from miRNAs. Although the study might be interesting for people in this field however there are several concerns that need to be clarified:
1) For this study HCT-116 and SW480 CRC cell lines has been used. There are several CRC cell lines known till date. The reason for choosing these cell lines must be justified. Moreover the reason for choosing two cell lines specifically must be clarified.
2) There are several stages and subtypes of CRC. Does the use of this monotherapy or triple therapy exhibit any effect on stage specific/subtype specific CRC. Do these cell lines represent any specific stage/subtype of CRC?
3) How were the dosages for monotherapy or triple therapy determined must be mentioned clearly.
4) As mentioned by authors 'miR-106 was detected as the best candidate from Targetscan'. How do authors determine miR-106 as best candidate must be mentioned. Did they use any specific criteria i.e. score/binding efficiency to determine miR-106 as the best candidate?
5) Authors mention that miR-106 is known to be highly expressed in colorectal tumor as compared to normal. Authors must show qPCR results for the expression of miR-106 in case of HCT-116 and SW480 CRC cell lines as compared to normal. Also it is important to show the expression of ULK1 in CRC cell lines as compared to normal.
6) Experiments in this manuscript demonstrate that the use of triple therapy leads to autophagy due to the increase in expression of ULK1. It has also been shown that upon triple therapy, the expression of miR-106 goes down. What is the mechanism behind downregulation of miR-106 upon TT? What are the key players/factors that are regulating the expression of miR-106?
7) A miRNA can have multiple targets. It is quite possible that miR-106 might be targeting other mRNAs as well and that might lead to disruption of other pathways in vitro. How do authors ensure that miR-106 is not causing any other adverse effect in vitro by targeting any other key gene? Please comment.
8) There are literature reports suggesting that ULK1 can be targeted by miR-26a-5p in colorectal cancer. Please comment.
9) Authors have shown the entire regulation and the effect of triple therapy in cell lines. It is important to understand the regulation by the miRNA and the effect of the drug in primary tissue samples. Do the results correlate well in primary tissue samples as well?
Author Response
Response to Reviewer 1 Comments
Point 1. For this study HCT-116 and SW480 CRC cell lines has been used. There are several CRC cell lines known till date. The reason for choosing these cell lines must be justified. Moreover the reason for choosing two cell lines specifically must be clarified.
Response 1. SW480 (ATCC CCL-228) is a Dukes´type B colorectal adenocarcinoma from a human male staged II; HCT116 (ATCC CCL-247) is a colorectal carcinoma cell line from a human male staged IV i. est. metastatic disease. Both cell lines represent different tumor stages (doi:10.1016/j.ejcb.2017.07.003). Manuscript page 3, lines 105-108.
In this research, we employed two colon cancer cell lines belonging to different CRC clinical stages in order to compare the effect of the triple therapy in early and advanced stages. As seen in our results, the response to treatment is very similar for both colon cancer cell lines, indicating the possibility of the triple therapy to be used in different stages of the cancer progression.
Point 2. There are several stages and subtypes of CRC. Does the use of this monotherapy or triple therapy exhibit any effect on stage specific/subtype specific CRC. Do these cell lines represent any specific stage/subtype of CRC?
Response 2. According to the National Cancer Institute, USA, the CRC is classified into four main stages I-IV. Depending on the spread of the tumor is the stage. The SW480 cell line nature corresponds to stage II. This is an early stage where the tumor spreads out of the walls of the colon and could have invaded other tissues but the lymph nodes. The treatment includes surgery and adjuvant treatment is recommended to avoid disease recurrence. Thus, our triple therapy could be used in the early stages of cancer, but due to its mechanism of action, it might be also tested in the most advanced colon cancer stages. Meanwhile, HCT116 is metastatic as aforementioned.
3) How were the dosages for monotherapy or triple therapy determined must be mentioned clearly.
Response 3: The dosages used in this study were the IC50 values determined by the Sulforhodamine B protocol described in (doi:10.1038/nprot.2006.179) and cited in our previous research articles (doi: 10.7150/jca.13123; doi: 10.7150/jca.16387). Briefly, we describe for your convenience the methodology we used. Cell lines were incubated in 96-well plates at a density of 7000 cells/well with different concentrations of doxorubicin, metformin, or sodium oxamate. For day 0, in a separate plate were seeded the cells in the same density and allowed to attach for 4 hours. At the end of the incubation time, either with or without treatment, cells were fixed with cold TCA 10% at 4ºC for one hour. After that time, the TCA was removed and plates washed with tap water and left them to dry at room temperature. After that, 100 L of SFR-B were added to each well and left them at room temperature for 30 min. Cells were washed with 1% of acetic acid to remove unbound dye. The dye was dissolved in 10 mM Tris base solution (pH 10.5) and stirred over 5 min. Optical density was measured in an Epoch Microplate Spectrophotometer (Bio-Tek) at 510 nm. Results are expressed as the concentration that inhibits 50% of control growth after the incubation period (IC50). The values were estimated from a semi-log plot of the drug concentration (mM/mL and M/mL) against the percentage of growth inhibition. Individual treatments were performed with their respective IC50. For the triple therapy, we used the IC50 of each drug to recalculate a new IC50 in combination, taking as a start point the individual IC50. In the manuscript in page 3, lines 108-110 was included the next text: “The drugs used were Doxorubicin (Dox, 0.75 μM) and for triple therapy Metformin (Met, 12 mM), Sodium Oxamate (Ox, 10 mM), and Doxorubicin (Dox, 0.4 μM)”.
4) As mentioned by authors “miR-106 was detected as the best candidate from Targetscan”. How do authors determine miR-106 as best candidate must be mentioned. Did they use any specific criteria i.e. score/binding efficiency to determine miR-106 as the best candidate?
Response 4: We lament having used a simplistic word for even more complex analysis. The first characteristic that led us to consider mir-106a is that the seed region has a length of 8-mer; which means that the sequence is highly conserved. Next, we have employed the context ++ score algorithm implemented in targetscan, which primarily considers the site type, 3’-supplementary pairing, and distance to closest 3’UTR end, among others (https://elifesciences.org/articles/05005). Using this methodology, the interaction between miR-106a and ULK1 has a context++ score of -0.19, where a negative score is better. Such data earns miR-106a the third-best possible regulator of ULK1 according to the targetscan algorithm.
However, it is widely known that in-silico prediction algorithms have a high rate of false positives. In this case, it is recommended to implement additional methodologies to further confirm this possible interaction. Thus, we correlated the expression of miR-106a and ULK1 using the RNA-seq data from the TCGA Colon-Adenocarcinoma project (https://www.nature.com/articles/nature11252) using the Starbase platform, which confirmed a negative correlation between the two molecules and setting mir-106a as a possible negative ULK1 regulator. In the text, the sentence structure has been changed (lines 215-216).
5) Authors mention that miR-106 is known to be highly expressed in colorectal tumor as compared to normal. Authors must show qPCR results for the expression of miR-106 in case of HCT-116 and SW480 CRC cell lines as compared to normal. Also it is important to show the expression of ULK1 in CRC cell lines as compared to normal.
Response 5: Thank you for the observation. These data were added as supplementary figure 4 a) and b). The text was modified (lines 245-252) Mir-106a expression was measured by qPCR in HCT116, SW480, and CRL1790, CRC cell lines and non-tumor cells. We observed a significant increase in the expression of miR106a in both cell lines. As observed in Supplementary Figure 4a, HCT116 had a six-fold overexpression, and SW480 had a four-fold overexpression. Besides, we analyzed the expression levels of mRNA ULK1, which has been proposed as a direct target of miR-106a, and found no significant changes of ULK1 in HCT116. However, in SW480, ULK1 expression decreased significantly (Supplementary material Figure 4b).
Point 6. Experiments in this manuscript demonstrate that the use of triple therapy leads to autophagy due to the increase in expression of ULK1. It has also been shown that upon triple therapy, the expression of miR-106 goes down. What is the mechanism behind downregulation of miR-106 upon TT? What are the key players/factors that are regulating the expression of miR-106?
Response 6: Thank you very much for your comments. Actually, the answer is extremely complex. It is well demonstrated that different drugs can modify the global expression profile both at the messenger and microRNAs level; for example, recently Rizzo and colleagues demonstrated that doxorubicin is able to modify the expression profile of MDA-MB-231 (TNBC) cells under Short Term Starvation (STS) condition ( 29069757). Similarly, pancreatic cancer cells exposed to metformin significantly increased the expression levels of nine microRNAs, some of them involved in apoptosis ( 22245693).
MicroRNAs, as well as mRNAs, are transcribed by RNApol2, so general and specific transcription factors are needed for miRNA transcription to be activated. In this sense, different articles have shown the transcriptional activation of microRNAs by specific Transcription Factors. miR106a is activated by KLF4 in gastric cancer (31146975) although this fact could have little or no relation with the pharmacological model we use. A short sentence was included in the Discussion section (lines 339-341).
Point 7. A miRNA can have multiple targets. It is quite possible that miR-106 might be targeting other mRNAs as well and that might lead to disruption of other pathways in vitro. How do authors ensure that miR-106 is not causing any other adverse effect in vitro by targeting any other key gene? Please comment.
Response 7. As the reviewer pointed out, a single microRNA can negatively regulate hundreds of mRNAs. We are sure of the necessity to demonstrate the negative regulation of a single microRNA in a target gene to know the role of microRNAs in such regulation. Although they are reductionist approaches, is also needed to validate the function of specific microRNAs in different aspects of cancer biology. In the case of the work presented here, a goal arising from the treatment with the triple therapy was the reduction of ULK1. A plausible hypothesis for the reduction observed was due to the participation of a microRNA. Certainly, miR-106a negatively regulates different genes in the tested cell lines, but it is not possible from our perspective, to assure that the role of this microRNA is only the regulation of ULK1.
Point 8. There are literature reports suggesting that ULK1 can be targeted by miR-26a-5p in colorectal cancer. Please comment.
Response 8. Several studies have reported that downregulation of ULK1 through different microRNAs could inhibit the progression of the autophagy process in many types of cancer (doi:10.1038/srep00808) (doi:10.1016/j.cellsig.2012.07.001) (doi:10.1158/1541-7786.MCR-17-0634). Among these microRNAs is mir-26a-5p, which can inhibit autophagy through the interaction of the ULK1 / ULK2 mRNA (doi:10.1038/s41598-018-20561-4) (doi:10.1111/rda.13177). For example, in human osteosarcoma miR-26a-5p, was found to regulate the autophagic pathway targeting ULK1 (doi:10.1186/s12935-019-0794-1). In prostate cancer cells, autophagy increases due to the overexpression of ULK1 / 2, but cells got resistant to different drug treatments, and when mir-26a was overexpressed, these cells become sensible (doi:10.1016/j.bbrc.2016.02.093). Likewise, with the information previously given, it is assumed that ULK1 is a direct target of mir-26a-5p in different types of cancer, however, so far in colorectal cancer, this information is scarce. Other report mentioned that SNHG6 promotes CRC cell autophagy through regulation of ULK1 via sponging miR-26a-5p and that miR-26a-5p suppresses ULK1 induced autophagy (https://doi.org/10.1186/s12935-019-0951-6). For this reason, it is important to study the regulation axis of microRNAs / autophagy in colorectal cancer.
Point 9. Authors have shown the entire regulation and the effect of triple therapy in cell lines. It is important to understand the regulation by the miRNA and the effect of the drug in primary tissue samples. Do the results correlate well in primary tissue samples as well?
Response 9. Thank you for pointing it out. However, testing the triple therapy in tissue samples was not the objective of this research. Since our results in commercial cell lines are very encouraging, the next step will be in primary cultures derived from patients’ samples, including early and late stages of the disease.
Final comment.
The abstract was modified and improved and the whole text was checked.

Reviewer 2 Report
In the present paper, the authors tested their previously validated triple pharmacological combination of metformin, doxorubicin, and sodium oxamate (triple therapy or TT) as an inducer of autophagy. The authors used the TT to induce autophagy in two cancer cell models and tried to identify the mechanism by which TT promote cell death and autophagy by monitoring the expression levels of miR-106a and its target ULK1. Although the scientific interest of the topic, the paper rises several major concerns:
-What is the rational for revalidating miR-106a as a regulator of ULK1 while its is already well validated? I don't see the part of luciferase assays necessary. Furthermore, the selection of miR-106a in particular should be justified since the beginning. The use of the expression "the best candidate
that could target ULK1" (line 210) seems not well founded and not appropriate.
-According to microRNA databases such as Target scan, miR-106a has two binding sites in human ULK1 3'UTR; Position 475-482 of ULK1 3' UTR and Position 771-777 of ULK1 3' UTR respectively. Could the authors justify why they have tested only one site?
- Exploring the autophagic/apoptotic effect of TT seems redundant with the previously reported work of the authors in which they have explored this triple pharmacological combination. I would rather suggest to further focus on the mechanisms than revalidated what is already established.
-Line 236: "As clearly seen in Figure 3d and 3e, we can deduce that ULK1 is not a mir106a target." Could you please further explain how have you arrived to this conclusion specially that both cell lines are transfected with miR-106a binding site at the ULK1 3'UTR region? hypothesis are required here.
-It is well established that Lipofectamin 2000 mediated transfection lead to an increased autophagy and apoptosis. How could the authors correlate the observed autophagy and apoptosis in their cells models to their experimental condition only (TT and miR-106a)?
Minor comments:
- Line 220: "had a marked dropped....", Do you mean "had a marked decline..."?
I would highly recommend the authors to reformulate their paper by summarizing the already established facts and putting central the new data such as the effect of TT on miR-106a/ULK1 expression. I would also suggest to deeper focus on mechanisms to add more value to the work.
Thank you
Author Response
Response to Reviewer 2 Comments
Point 1. What is the rational for revalidating miR-106a as a regulator of ULK1 while its is already well validated? I don't see the part of luciferase assays necessary.
Response: We appreciate your question, the purpose of conducting the luciferase reporter gene assay to demonstrate the binding between miR-106a and the 3' UTR region of ULK1 is because there is only one published evidence on lung cancer. We thought it was necessary to perform this experiment to corroborate the already published data (doi:10.1016/j.lungcan.2016.06.004.)
Point 2. Furthermore, the selection of miR-106a in particular should be justified since the beginning. The use of the expression "the best candidate that could target ULK1" (line 210) seems not well founded and not appropriate.
Response: We lament having used a simplistic word for an even more complex analysis. The first characteristic that led us to consider mir-106a is that the seed region has a length of 8-mer; which means that the sequence is highly conserved. Next, we have employed the context ++ score algorithm implemented in targetscan, which primarily consider the site type, 3’-supplementary pairing, and distance to closest 3’UTR end, among others (https://elifesciences.org/articles/05005). Using this methodology, the interaction between miR-106a and ULK1 has a context++ score of -0.19, where a more negative score is better, which makes miR-106a as the third-best possible regulator of ULK1 according to the targetscan algorithm.
However, it is widely known that in-silico prediction algorithms have a high rate of false positives it is recommended to implement additional methodologies to further confirm this possible interaction. Thus, we correlated the expression of miR-106a and ULK1 using the RNA-seq data from the TCGA Colon-Adenocarcinoma project (https://www.nature.com/articles/nature11252) using the Starbase platform, which confirmed a negative correlation between the two molecules and putting mir-106a as a possible negative ULK1 regulator. In the text the sentence structure has been changed (lines 215-216).
Point 2. -According to microRNA databases such as Target scan, miR-106a has two binding sites in human ULK1 3'UTR; Position 475-482 of ULK1 3' UTR and Position 771-777 of ULK1 3' UTR respectively. Could the authors justify why they have tested only one site?
Response: Dear reviewer, according to the targetscan algorithm, the position 771-777 is poorly conserved, whereas the other site which we validated is highly conserved, in addition the position 771-777 has a relatively bad context score (-0.02). Taking in account this, it was highly more probable that if the interaction between the two molecules existed, it would be given by the 475-482 region due to all of its characteristics evaluated by the targetscan algorithm.
Point 3. Exploring the autophagic/apoptotic effect of TT seems redundant with the previously reported work of the authors in which they have explored this triple pharmacological combination. I would rather suggest to further focus on the mechanisms than revalidated what is already established.
Response: thank you for your observation. As you mentioned, we have previous work with the TT in breast and colon cancer, both of them with evidence in vitro and in vivo of the apoptotic effect of our combination and the induction of autophagy. In this work, we continue searching for the mechanism of action of the TT specially as a pharmacologic inducer of autophagy. Our findings showed that TT decreased miR-106a expression in CRC cells, which leads to the induction of the autophagy process and subsequently apoptosis. The evidence here presented strengthens the anti-tumor action of the combination of the three drugs as inducer of autophagy and apoptosis. Further research should be done to understand and unravel its potential as a cancer treatment.
Point 4. Line 236: "As clearly seen in Figure 3d and 3e, we can deduce that ULK1 is not a mir106a target." Could you please further explain how have you arrived to this conclusion specially that both cell lines are transfected with miR-106a binding site at the ULK1 3'UTR region? hypothesis are required here.
Response: Thank you for your key observation. We have changed the sentence to make it clear. We improved as follows ”As clearly seen in Figure 3d and 3e, luminescence was not reduced in co-transfected cells, suggesting that the effect is cellular context dependent.” Lines 241-241.
Point 5. It is well established that Lipofectamin 2000 mediated transfection lead to an increased autophagy and apoptosis. How could the authors correlate the observed autophagy and apoptosis in their cells models to their experimental condition only (TT and miR-106a)?
Response: In this work we performed all the transfections using lipofectamine 2000 as previously described for these cell lines (DOI: 10.29252/mlj.12.4.12).
Although, previously it has been reported that the use of lipofectamine 2000 allows the increase of autophagy and apoptosis pathways because it is a cationic lipid (DOI: 10.4161/auto.6.4.11612), the activation of these cellular processes has been related in function of the concentration and time of exposure to lipofectamine (doi: 10.1007/s12033-011-9422-6). So, in order to avoid an increase in autophagy and apoptosis, we use half the concentration suggested by the manufacturer; which did not impact transfection efficiency because it has already been reported good results even using 1 µl of Lipofectamine 2000 in transfection assays (DOI: 10.29252/mlj.12.4.12). We demonstrate this event with the transfectant cells that only express GFP-LC3 because we did not observe the expression of the green fluorescent protein, so we can suggest that at least in this case there is no increase in the autophagy pathway due to the use of lipofectamine 2000. That is why our results with the experimental conditions evidence the activation of the autophagy pathway as a function of triple therapy and miR-106a.
Minor comments:
- Line 220: "had a marked dropped....", Do you mean "had a marked decline..."?
Response: We have changed the sentence as suggested
I would highly recommend the authors to reformulate their paper by summarizing the already established facts and putting central the new data such as the effect of TT on miR-106a/ULK1 expression. I would also suggest to deeper focus on mechanisms to add more value to the work.
Response: Dear Reviewer,
We value your comments and suggestions to enrich this article. However, we consider that the structure of the work is properly discussed according to our findings. Moreover, the mechanisms involved in the overall results are still to be understood. We below summarize it for your consideration:
Our research group proposed several years ago a therapy with the combination of three known drugs, two antineoplastic drugs that are doxorubicin and metformin, and sodium oxamate. In those studies, we observed that the named triple therapy has the ability to induce apoptosis and autophagy in breast and colon tumor cells. Continuing with the search for the mechanism of action of these drugs, in this work we look for their possible target for the induction of autophagy as cell death in colon cancer tumor cells. Firstly, through WB trials we observe that after 8 hours of stimulation with the three drugs, the proteins ULK1, becline1, Atg4, and LC3 increase in the cell lines HCT116 and SW480 concerning doxorubicin monotherapy. The overexpression of these proteins indicates the beginning of autophagy flow through the activation of ULK1 as the initiator protein of this process and the hyperlipidation of LC3. Likewise, while ULK1 is overexpressed, miR106a expression declines when cells are treated with triple therapy, while doxorubicin has the opposite effect. We were able to show that miR-106a negatively regulates ULK1. Afterward, through microscopy, we could observe that the formation of autophagosomes was indeed taking place in those cells treated with the triple therapy.
This work provides, through different techniques, evidence about pharmacologically induced autophagy in CRC cell lines However, this work is not conclusive and it is required research that allows us to understand other mechanisms of action and metabolic pathways that turn on or off with this therapy. We believe that the structure of the manuscript is presented logically; thus its lecture and analysis are easy for the reader.
Final comment.
The abstract was modified and improved and the whole text was checked.

Round 2
Reviewer 1 Report
Comment to 5: As no significant change in expression for ULK1 was found in HCT116 as compared to the control cell line, how can authors consider ULK1 as a bonafide target of miR-106 in HCT116 cell line. Please justify.
Comment to 7: Authors must add few lines corresponding to this in the manuscript.
Author Response
Reviewer 1
Comment to 5: As no significant change in expression for ULK1 was found in HCT116 as compared to the control cell line, how can authors consider ULK1 as a bonafide target of miR-106 in HCT116 cell line. Please justify.
Response: In the current work, we found overexpression of mir-106a and significant downregulation of ULK1 in SW480 (supplementary figure 4a and b), showing a negative correlation between mir-106a and ULK1, suggesting a possible axis regulation in this cell line. Nevertheless, in HCT116 cells, whereas the expression of both molecules did not seem to have a correlation, the experimental data showed a significant level of correlation with the prediction algorithms. Thus the validation of that was shown by luciferase activity reduction and ULK1 protein decrease (Figure 3b and 3c).
miRNAs are non-coding short RNAs that target the 3’UTR of mRNAs, whereas AGO proteins function as effectors by recruiting factors that induce mRNA de-adenylation, mRNA degradation, or translational repression without affect mRNA target levels (doi.org/10.1016/S0092-8674(04)00045-5). Recent studies indicate that GW182 family proteins interact directly with AGOs; GW182 competes with eIF4G in association with a poly-A binding protein, preventing the circularization required for efficient translation (doi.org/10.1186/1758-907X-1-11). In this way miRNAs could regulate translational repression of target mRNA, which explains that despite not showing significant changes in the expression of ULK1 in HCT116, the regulation exerted by mir-106 could be leading at the protein level, a fact demonstrated by western blot in figure 4c.
The relation between mir-106a and ULK1 has been widely described in other cancers (doi.org/10.3389/fimmu.2020.610021) (doi.org/10.1016/j.lungcan.2016.06.004), evidencing the importance of this axis of regulation in the autophagy activation; added to this and to the data obtained experimentally, we affirm that ULK1 is a genuine target of mir-106a in CRC.
Comment to 7: Authors must add few lines corresponding to this in the manuscript.
Response: Thank you for the suggestion. We have included the response into the text. Page 11, lines 346-354.
Reviewer 2 Report
I thank the authors for addressing most of the first review round commentaries. However, the paper still presents some flaws particularly related to the experimental designs and statistical data.
Major comments:
-The Luciferase assays: The authors should discard the hypothesis that the absence of miR-106a effect in SW480 cells is not due to a very low levels of endogenous (basal) of ULK1 mRNA in these cell lines. It is necessary to measure the basal (endogenous) expression levels of ULK1, miR-106a and miR-1 (your control) in your cell lines. This would avoid any misinterpretation of your expression data.
-Key proteins densitometry data after the different treatment conditions: The authors refer to the supplementary figure S1 when speaking about the "evident" difference of protein expression between the TT versus doxorubicin. However, there is no concordance between the huge error bars and the statistical significance of the data in most of the panels:
- Panel a) ATG4 after 12h and 24h of treatment with Doxorubicin.
- Panel b) LC3II/I (+), after 4h and 6h of TT
- Panel c) ULK1 after 6h, 8h, 12h and 24h of Doxorubicin treatment.
- Panel d) ATG4 after 24h of TT, LC3II/I after 4h of TT
-Autophagy testing: I would suggest to the authors to include another internal control which is cells treated only with Lipofectamin 2000 in order to normalize the autophagic effect of Lipofectamin 2000. When justifying the protocol, the authors refer to a published paper (DOI: 10.29252/mlj.12.4.12) where no autophagy rate was tested and where 1µl of Lipofectamin for a 24h exposure (as used in the present paper) is judged not efficient ("At 24 hours post-transfection, the cells transfected with 1 µl Lipofectamine and different amounts of plasmid did not show any significant fluorescence activity." DOI: 10.29252/mlj.12.4.12)
Minor comments:
Please include the arguments (summarized) provided to justify the miR-106a choice in the manuscript so the readers could understand the rational (Line 215-216).
Thank you.
Author Response
-The Luciferase assays: The authors should discard the hypothesis that the absence of miR-106a effect in SW480 cells is not due to a very low levels of endogenous (basal) of ULK1 mRNA in these cell lines. It is necessary to measure the basal (endogenous) expression levels of ULK1, miR-106a and miR-1 (your control) in your cell lines. This would avoid any misinterpretation of your expression data.
Response: Dear reviewer, the endogenous levels of ULK1 and mir-106a was measured in the cell lines (supplementary figure 4a and b). We did not measure mir-1 levels due to its overexpression by mimic transfection does not exert a significant effect on mir-106a expression in both cell lines as we mentioned in supplementary figure 4c and d. In lines 241-243 we mention that the absence of mir-106a effect on ULK1 in SW480 is due to cellular context and not low levels of endogenous ULK1 mRNA. In spite of we transfected the same reporter luciferase plasmid and the same mimic concentration in both cells, the luciferase activity was totally different, indicating an evident response mediated by cellular context. So another mechanism was switched by mir-106a independent of ULK1 in SW480 since autophagy was activated as showed in Figure 4b. However, the mechanism is still unknown to us and it is impossible to clarify for the moment with our methodological strategy, but it does not underestimate the effect that triple therapy has on the activation of autophagy-mediated by this miRNA.
-Key proteins densitometry data after the different treatment conditions: The authors refer to the supplementary figure S1 when speaking about the "evident" difference of protein expression between the TT versus doxorubicin. However, there is no concordance between the huge error bars and the statistical significance of the data in most of the panels:
- Panel a) ATG4 after 12h and 24h of treatment with Doxorubicin.
- Panel b) LC3II/I (+), after 4h and 6h of TT
- Panel c) ULK1 after 6h, 8h, 12h and 24h of Doxorubicin treatment.
- Panel d) ATG4 after 24h of TT, LC3II/I after 4h of TT
Response: thanks for the observation. We had realized about the huge error bars, and while the manuscript was revised, we performed some experiments to have more representative graphs and thus increase the sample number. Therefore, the standard deviations here showed are lower (four independent experiments). Moreover, densitometric analysis was modified and distributed into two sheets so that the difference among the treatments could be better observed (supplementary material Figure 1).
-Autophagy testing: I would suggest to the authors to include another internal control which is cells treated only with Lipofectamin 2000 in order to normalize the autophagic effect of Lipofectamin 2000. When justifying the protocol, the authors refer to a published paper (DOI: 10.29252/mlj.12.4.12) where no autophagy rate was tested and where 1µl of Lipofectamin for a 24h exposure (as used in the present paper) is judged not efficient ("At 24 hours post-transfection, the cells transfected with 1 µl Lipofectamine and different amounts of plasmid did not show any significant fluorescence activity." DOI: 10.29252/mlj.12.4.12)
Response: Dear reviewer, we did not include the result of the lipofectamine as another control in the original manuscript since we didn't consider it necessary; however, we followed your suggestion and added it to Figure 4 (labeled: GFP-LC Lipofectamine). since we did perform that experiment previously. Page 4, line 171 and page 6 lines 270-271.